# Inclusion and Inclusive Education in Russia: Analysis of Legislative and Strategic Documents at the State Level between 2012–2014

**Natallia Bahdanovich Hanssen** [1,*] and **Aleksandra A. Alekseeva** [2]

1 Faculty of Education and Arts, Nord University, 8049 Bodø, Norway
2 Independent Researcher, Moscow 117279, Russia; alekswiz@gmail.com
* Correspondence: natallia.b.hanssen@nord.no

**Abstract:** Inclusive education is an important foundation of many societies, including the post-Soviet countries. It has been more than ten years since the official implementation of inclusion in Russia. However, the inclusive education system has not developed enough to be equally supported everywhere throughout the country, and is marked by controversial views on legal regulation and inclusive strategies. The purpose of this article is to examine inclusion and inclusive education, mainly for students with special educational needs, as understood in the educational policy. The data consist of legislative and strategic documents on the state level between 2012 to 2014. The data analysis was based on a qualitative content analysis. The analysis indicated two main themes: the subtle expression and lack of a definition for inclusion, and an inconsistent expression and definition of inclusive education. The results point to the necessity of encouraging discussions as well as reflections with respect to articulating and defining what inclusion is and how Russia can create an effective strategy for the further development of inclusive education.

**Keywords:** inclusion; inclusive education; QCA; Russia; special needs education; legislative and strategic documents at the state level

## 1. Introduction

The vision of inclusion and inclusive education has been a guiding principle in many countries [1–3]. This international trend toward inclusion and inclusive education, especially for students with special educational needs (SEN), has also been evident over the last two decades in many post-Soviet countries, among them Belarus [4] Lithuania [5], Moldova [6], and Ukraine [7]. In these countries, the phenomena of inclusion and inclusive education have been studied in various ways and research in the literature claims that the understanding of inclusion and inclusive education is vague, tensional, confusing, contradictory, and closely linked to integration, focusing only on students with SEN and their process of assimilation into a predefined educational structure. Moreover, academics assume that a fundamental shift with respect to the way in which inclusion is being addressed, as education for all has not happened yet [4,5,7].

However, [8] highlights that, in post-Soviet countries since the beginning of XXI century, an initial move has been made towards an understanding of inclusion and inclusive education in their broadest sense, as values of human diversity, pluralism and respect for human beings independent of race, gender, disability or medical or other needs.

The Russian Federation has also slowly moved forward toward inclusion and inclusive education, and for example, ratified the Convention on the Rights of Persons with Disabilities [9] and legalised the right to inclusive education through its adoption of the federal law "On Education in the Russian Federation" No. 273-FZ of 29 December 2012 (FLE) in 2012 [10]. According to the FLE [10], all students should be provided equal access to education, especially considering the diversity of SEN and individual opportunities.

It has been twelve years since the implementation of inclusion, but equal educational opportunities for all, especially for students with SEN, seems to not yet have been sufficiently provided in Russia [11,12].

To boot, several researchers associated the failure to achieve the goals of inclusion with a misunderstanding and non-acceptance of students with SEN by teachers and by society in general [11–14].

It has also been reported that the main approved legislative and strategic documents have identified mechanisms for the implementation of inclusion and inclusive education, especially for students with SEN, but that these mechanisms are rarely enforced or fulfilled [12,15–17].

However, it is important to stress that the Russian Federation is a large country with high regional heterogeneity. The realisation of inclusion has had a rather diverse implementation across the country.

Rytova et al. [18] show that inclusive development does not always go hand-in-hand in Russian regions. Authors point out that Moscow ranks high in terms of inclusive development, while Tyva is one of the less-inclusive regions. The variations regarding inclusive development may also vary between cities within regions. In line with the critics of the UNESCO report [3], some researchers have pointed out that, in big cities such as Moscow or Saint-Petersburg, mainstream schools that are willing to enrol students with SEN do receive more additional funding than that received by small cities and villages within the same regions. The number of teachers with competencies in inclusion also prevail in big cities [3,19,20].

The given overview characterising the variation in achieving the goals of inclusion policy in Russia are somehow incomplete. Still, this overview sheds light on the concerns and the complexity of the elaboration of inclusion and inclusive education on the policy level. Therefore, it is surprising that there is a lack of studies clarifying the main legal and legislative challenges that have occurred in Russian policy regarding inclusion and inclusive education.

At the same time, there is a great need for a more in-depth understanding and clarification of the political aims and values that Russia has for the realisation of inclusion in an educational context [12]. Responding to this demand, the current article aims to answer the following question: *how is inclusion and inclusive education for students with SEN understood in the legislative and strategic documents at the state level between 2012–2014?*

In this article, the focus is on the legislative and strategic documents at the state level. We focus on an analysis of documents that were developed between 2012 to 2014. As can be seen, the documents chosen cover a very narrow (three-year) period, and do not cover the entire history of inclusion policy in Russia. Moreover, the documents chosen do not cover a number of local regulations in different regions. However, the documents have played a determining role in the development of inclusion and inclusive education, used federally in Russian, and have mainly remained unchanged.

Striving to avoid ambiguity and confusion over the terminology, and to find lenses that make it possible to answer the research question in the current article in an appropriate manner, it is essential to clearly define concepts of inclusion, inclusive education, and students with SEN.

Inclusion, as pointed out by Thomas [21], is a humanistic ethos that values human diversity and emphasises social justice and equity in society. At the same time, inclusion involves a particular emphasis on those groups of learners who may be at risk of marginalisation, exclusion, or underachievement [22]. Students with SEN can be described as a group of learners with a high risk of marginalisation. Therefore, we focus especially on inclusion for this group of students and follow Ainscow, Booth and Dyson [23] in their definition of inclusion as a principled approach 'concerned with children and young people [with SEN] in schools; it is focused on [their] presence, participation and achievement' [23].

Based on the narrow perspective of inclusion, inclusive education can be defined as providing access and participation, and meeting diverse individual learning needs in mainstream settings [24].

In Russia, various terms are used to describe students with SEN. According to Federal Law No. 181-FZ of 24 November 1995 [In Russian: Federalnij zakon o socialnoj zachite invalidov v Rossijskoj Federacii] "On the Social Protection of the [invalid] People in the Russian Federation" (Art. 1) [25], "*Invalid*" is a person who has health disorders with a persistent disorder of body functions, caused by diseases, the consequences of injuries or declines, leading to a limitation of life and causing the need for his social protection". Children-invalids in this document are referred to as those with the same range of disorders whose age is under 18 years old.

The term [In Russian: deti s ogranichennimi vozmoznostami zdorovja (OVZ)], children with limited health conditions (LHC) refer to individuals who have deficiencies in physical and (or) psychological development that hinder them from education without creating special conditions [10]. These deficiencies need to be confirmed by the Psychological-Medical-Pedagogical Commission.

From the humanitarian and ethical point of view, the term students with limited health conditions (LHC) would be most correctly used when directly citing the documents used.

However, the current article uses the term students with special educational needs (SEN) to designate both terms and applies to individuals whose age is under 18 years old.

## 2. Russian Educational Context

Educational policy is grounded on the creation and improvement of a high-quality national education system. Therefore, the development of education is among the state's priorities [10].

The first sparks of inclusion in Russia on non-governmental level with no government support happen in the 20th century [17]. By 2012, Russia officially started implementation of inclusion and transformation of the education system at the legislative level. First, Russian authorities signed in 2008 and then in 2012 ratified the Convention on the Rights of Persons with Disabilities. Then, the country approved the National Action Strategy for Children in 2012–2017 by the Decree of the President of the Russian Federation, dated 1 June 2012 (No. 76) [26].

To date, the realisation of the inclusive education is established in the main document that regulates all education in Russia, including the SEN structure, the FLE [10].

The FLE highlights the right to inclusive and antidiscriminatory education for all, including students with SEN (Art. 2; Art. 5) [10]. As such, the term inclusive education, in contrast to the term inclusion, has received normative consolidation and has been recognised as an overarching principle in education [11]. It is important to add that Russia is a federal state and a large country, and at the region and at the city level, there are multiple approaches towards inclusive education. For example, in various regions, education for students with SEN can be provided in inclusive education settings, in separate classes and groups, and via home-based education [3].

As pointed out by UNESCO [3], a successful move towards inclusion presupposes ensuring students' rights to education. In Russia, the right to special needs education (SNE) for students with SEN is statutory and regulated by the MESRF (Art. 42) [10]. Multiple government services and professionals from the fields of medicine, pedagogy, and psychology, via the Psychological–Medical–Pedagogical Commission (PMPC), assess each student's needs and provides recommendations on which educational programme and education placement (special or boarding schools or special classes) are best suited to each student (Art. 44) [10]. Still, the needs identification systems in big cities, such as, for example, Moscow, have been reorganised to be more inclusive than those in other regions [3]. However, according to UNESCO [3], a system depending on legalised selection procedures can be assumed to be a barrier hampering equal and fair school access.

Furthermore, for the enhancement of the positive positioning of inclusive education in society, numerous associations, networks, and non-governmental organisations (NGOs) have been formed to monitor legislation implementation as well as to press for inclusive education [3,11]. At the same time, in many cases, the voices of NGOs have not even been taken into consideration regarding the development and promotion of inclusion in education [3].

Nonetheless, Russia initiated a movement towards the minimisation of a number of special schools. Guided by the state programme, Accessible Environment 2011–2015, several actions were undertaken in the country, such as improving mainstream schools' physical accessibility, and adapting environments to appeal to all users, including those with SEN [3,11,27]. However, UNESCO [3] states that, for all these years, Russia has not promoted efficient resource use, and not all mainstream schools receive additional funds to enrol students with SEN. Despite several positive efforts towards inclusion, the educational system remains largely unchanged [3,8,14]. According to the Federal State Statistics Service (Rosstat), there were 729,000 students with SEN under the age of 18 in 2019. This number includes preschool and school-aged individuals. The same source states that there were 31,589,775 children under the age of 18 in Russia in 2019 overall (making it 2.3% of children being children with SEN). A total of 40,000 students with SEN are educated at home (5.49% of all students with SEN), and about 70,000 (9.6% of all students with SEN) are educated at special schools or boarding schools, being limited to learning alongside their peers in local community schools [28].

## 3. Method

In this article, the focus is on legislative and strategic documents at the state level. We want to explore how the concept of inclusion and inclusive education for students with SEN is understood in legislative and strategic documents at the state level.

### 3.1. Document Selection

To obtain relevant information, general criteria for the selection of documents were established, including that the documents should have a determining role in the development of inclusion and inclusive education in Russia, especially for students with SEN. Another criterion was representativeness concerning the applicability of the documents throughout the country. It was determined that the chosen documents should be used federally, applied everywhere, and should have the greatest amount of influence. As such, we focused on the legislative and strategic documents at the state level that were issued between 2012 and 2014 and that are still eligible. Following these criteria, a total of four main legislative and strategic documents at the state level were chosen. All editions of the chosen documents were taken into consideration. In this sense, the study must be recognised as an explorative study.

The documents in the analysis are presented in chronological order (see Table 1):

**Table 1.** Analysed documents.

| | Title in English | Title in Russian | Referred to as | Description |
|---|---|---|---|---|
| 1 | Decree of the President of the Russian Federation of 1 June 2012 No. 761 "On the National Strategy for Action in the Interests of Children for 2012–2017" | Указ Президента Российской Федерации №761 от 1 июня 2012 г. «О национальной стратегии действий в интересах детей на 2012–2017 годы» | The Strategy, 2012 [26] | The national Strategy is a document that summed up the endeavors to protect children's rights in the Russian Federation through legal documents before the federal law "On Education in the Russian Federation" was established. Following the Russian Constitution's guarantee of the state's support for the family, motherhood and childhood; signing the Convention on the Rights of the Child [29], the Strategy reflects the commitment to participate in the efforts of the world community to create an environment that is comfortable and friendly for children. It contains an analysis of problematic aspects of childhood in the region and actions that are expected to be done to resolve these issues. |

**Table 1.** *Cont.*

| | Title in English | Title in Russian | Referred to as | Description |
|---|---|---|---|---|
| 2 | Federal Law "On Education in the Russian Federation" No. 273-FZ of 29 December 2012 | Федеральный закон "Об образовании в Российской Федерации" от 29 декабря 2012 г., №273-ФЗ | FLE, MESRF, 2012 | A fundamental document that sets the legal basis for regulating the sphere of general education in the Russian Federation. FLE is the first document which is aimed to give an official right to inclusive education in Russia as well as the first legal document stating parents' right to choose where their child can receive their education (what kind of school, and any school must accept their will). |
| 3 | The Order of the Ministry of Education and Science of the Russian Federation No. 1598 dated 19 December 2014. "On the approval of the Federal State Educational Standard of Primary General Education for Students with limited health conditions | Федеральный государственный образовательный стандартначального общего образования обучающихся с ограниченными возможностями здоровья (утв. приказом Министерства образования и науки РФ от 19 декабря 2014 г. N 1598) | FSES for Primary General Education, MESRF, 2014 [30] | FSES for Primary General education, MESRF, 2014 includes such disabilities as deaf, hard of hearing, late deaf, blind, visually impaired, with severe speech impairments, disorders of the musculoskeletal system, mental decline, autistic spectrum disorders, and severe defects. |
| 4 | The Order of the Ministry of Education and Science of the Russian Federation No. 1599 of 19 December 2014. "On the approval of the Federal State Educational Standard for the Education of Students with Intellectual Disabilities | Федеральный государственный образовательный стандартобразования обучающихся с умственной отсталостью (интеллектуальными нарушениями)" (утв. приказом Министерства образования и науки РФ от 19 декабря 2014 г. N 1599) | FSES for the Education of students with ID, MESRF, 2014a [31] | It comprises compulsory educational requirements. The Federal State Educational Standard establishes qualitative and quantitative criteria in education, like standards in sports or products. It is used as a practical tool to support educational organisations in their everyday relations with students with SEN and their families. The FSES for the Education of students with ID, MESRF, 2014a includes such health conditions as mild mental decline (intellectual disabilities), moderate, severe, profound mental decline (intellectual disabilities), and severe and multiple developmental disabilities. |

*3.2. Analysis*

In the current study, the data consisted of texts. The texts were analysed with the help of a qualitative content analysis (QCA). QCA has a long history, particularly in nursing research. It refers to systematically describing the meaning of qualitative material. QCA is most frequently applied to verbal data, but QCA can also be used for analysing textual data. It selects key aspects that researchers want to pay attention to, and develops and aggregates categories in order to grasp their meaning [5,32,33]. Analysing the content of the legislative and strategic documents at the state level with help of QCA the process was guided by theoretical perspective revealing inclusion and inclusive education. Therefore, the main concepts of *inclusion* and *inclusive education* were used as the preliminary frames of categorisation. After the chosen frames of categorisation, data were reviewed for content and coded for correspondence with the identified categories [32]. The process was as follows: authors first independently familiarised themselves with the data. The data were thoroughly read word by word, sentence by sentence. The impressions from the reading resulted in a very large number of codes that did not always correspond with the preliminary frames of categorisation (for example integration, included in class, accessibility, socialisation, equal opportunities, choice, adaptation, elimination of the barriers, separate groups, educational programs, combat discrimination, quality of education, correction of disorders, etc.). Therefore, the creation of extended categorisation frames and a greater degree of reduction was demanded. The next step was to systematically search for focus categories that were more consistent than the codes. Each author sorted the codes by comparing them for similarities and differences and grouping those with similar meanings into the predefined frames of categorisation. This process reduced the material to a manageable format. Finally, by following the same process again, authors abstracted focus categories into main categories of description related closely to the data [33]. After this individual process, a collective agreement was developed about the data. The process was similar to the previous process; by systematically comparing and recognising the coherence between the codes, authors gathered them into more complex focus categories, which were descriptive, and also repeated the documents' statements. Then, the focus categories were compared for

similarities and differences, and when they shared a similar meaning, they were grouped into the same subcategories. The subcategories were then reviewed and modified. Finally, by following the same process again, the dominant features of the common main categories were developed. The main categories were *the subtle expression and lack of a definition for inclusion* and *an inconsistent expression and definition for inclusive education.*

The analysis process that led to the constitution of the identified categories emerged as obvious, unproblematic and clear when seen from one side. However, our analysis process is value-laden and attached to our frame of references, and may appear as too unilateral by providing an illusory view of exactness [34,35]. Semantically, the selected categories may also refer to various meanings, depending on researchers' different cultural and political conditions. Regardless, the identified categories strive to capture features valid for conditions in a respective country [34]. Despite these critical remarks, the categories express a profiled densification of characteristics and obvious features, thereby exposing an understanding of inclusion and inclusive education for students with SEN in the legislative and strategic documents at the state level.

## 4. Limitations

Several limitations should be taken into consideration. First, it is central to stress the fact that, due to limited access to the scientific resources, the picture of the contextual situation in Russia may appear as limited and less nuanced. As such, the presentation and discussion of results in the study were impacted with respect to answering the research question of how inclusion and inclusive education for students with SEN is understood in the educational policy between 2012 and 2014.

Moreover, the small sample of documents could indicate that the results may have less validity or may be considered as broader generalisations. Further, the chosen documents are within the period 2012–2014, which is a very narrow (three-year) period and does not cover the entire history of inclusion policy in Russia. In addition, the documents analysed do not reveal interregional differences among Russian regions. However, those documents have a determining role in the ongoing development of inclusion and inclusive education at the federal level in Russia.

As far as possible, we tried to make the study reliable and valid by maximising authenticity, transparency, and honesty in all areas of the research process [36]. We strived to reach naturalistic generalisations with the aim of heightening the utility of the study's findings.

The study may be perceived as useful to the reader's own situation, where the reader can recognise their own situation through the findings. However, it is up to the reader to decide whether the results are beneficial and transferable to other contexts [33,37].

Regarding the analysis itself, the use of QCA and delineation of two main categories can be interpreted as rather superficial. However, delineation, on the one hand, helped us to reduce the diversity of meanings in the data to the distinctions specified by two categories. On the other hand, the data not covered by these components was lost for further analysis [33,38]. We assume that some interesting themes that seem evident from the excerpts presented in findings section, for example that documents seemed to be multiple and offer contradictory ways to understand inclusion and inclusive education (e.g., integration, segregation, accommodation, parental freedom to choose schools). These additional themes were pointed out to some extent in the discussion but were not explicitly outlined as separate themes in the results. However, the main categories chosen were considered appropriate with respect to the research question in the article.

Finally, there is a limitation connected to the conceptualisation of inclusion and inclusive education as applied in the current article. As previously stated, to create a meaningful study, the research question was to direct the reader's attention to understanding of inclusion and inclusive education for students with SEN in the legislative and strategic documents at the state level between 2012–2014.

This is a challenging task, especially when dealing with heterogeneous phenomena of inclusion and inclusive education, which are strongly influenced by, i.e., history, culture, and social context [36].

Researchers can address these limitations by carrying out similar research and analysing several legislative and strategic documents at the state level to provide a more accurate understanding of inclusion and inclusive education for students with SEN in Russia.

## 5. Results

The findings are organised according to the identified two main themes and described through excerpts from the documents analysed.

### 5.1. The Subtle Expression and Lack of a Definition for Inclusion

The first theme focuses on the ways that the concept of inclusion is expressed and defined in the analysed documents.

Despite the fact that the legislative and strategic documents are aimed at developing and implementing inclusion, clear or direct references to the word inclusion were not found in any of the analysed documents. We could not find a clear definition for inclusion in any of the analysed documents.

For example, the Order of the Ministry of Education and Science of the Russian Federation No. 1599 of 19 December 2014, "On the approval of the Federal State Educational Standard for the Education of Students with Intellectual Disabilities" (FSES for the education of Students with ID) [31], does not mention inclusion at all. The Order of the Ministry of Education and Science of the Russian Federation No. 1598 dated 19 December 2014, "On the approval of the Federal State Educational Standard of Primary General Education for Students with limited health conditions" (FSES of Primary General Education) [30], at least mentions inclusion seven times. However, the term inclusion in this document is used interchangeably with the term integration, aiming at incorporating students with SEN in educational system with help of correctional programs, as the main requirements for the provision of inclusion:

"Any correctional program should provide an opportunity for the students with LHC to master the educational program and their integration in the organization" (FSES of Primary General Education, Appendix 1–8) [30].

In the same document, for example, inclusion is further mentioned as an integration, but this time is connected to the educational flow of the students with hearing impairments (HI) [30]. More precisely, the Appendix of FSES for Primary General Education [30] clearly expresses that inclusion does not guarantee all students with HI access to education, but only if certain demands are fulfilled: "students with HI can be included in class if there are no more than 2 of those in the class" (FSES of Primary General Education, Appendix 2, Point 2.1) [30]. Counterintuitively, MERSF [30] is subordinated to the Federal Law "On Education in the Russian Federation" No. 273-FZ of 29 December 2012 [10], which clearly states the exclusive parental right to choose educational institutions despite any limitations in the latter documents. In other words, the MERSF [30] is based on FLE [10], but we can assume that this document impedes the law in this matter.

The Decree of the President of the Russian Federation of 1 June 2012 No. 761 "On the National Strategy for Action in the Interests of Children for 2012–2017" [26] outlines the main aims regarding inclusion, but without mentioning directly what this concept or term means. The Strategy [26] emphasizes reformation of the network and activities of institutions for orphans and students left without parental care, including "for students with LHC of all kinds". The aim of the reformation is to make the system to become friendlier and more hospitable towards students and their families, either adoptive or biological, to include them in society. As the same time, the Strategy [26] demonstrates an indirect understanding of inclusion as giving the right to education for all, and highlights the idea of reorganisation of the ordinary educational environment in support of students with SEN:

"All kinds of support for families raising children with LHC: creating a modern comprehensive infrastructure for rehabilitation and educational assistance to students with LHC. Introducing these students into the environment of ordinary peers, ensuring their full socialization life in future adult life" [26].

When it comes to Section 5 of the Strategy [26], we can observe more direct mentions of inclusion, although this term is not named. The Strategy uses terms such as "equal opportunities', 'anti-discriminatory principle", "adaptation". The Strategy [26] highlights the possibility of common educating activities for students with SEN, by "definition of their abilities and possible educational paths". Furthermore, the Strategy [26] calls attention to the development of an overall awareness and the formation of a positive attitude towards students with SEN as equal members of society. The Strategy (Section 5) [26], further promotes the significance of responsible parenting through assuring a unified system of counselling assistance to parents.

Following the Strategy [26] further, the analysis shows that the document does not precisely define the groups of students that this inclusion may concern, as the FSES for the Education of students with ID [31] does. As such, inclusion may seem to apply to *all* students, including those in orphanages, students with various disabilities, students living on the edge of poverty, social orphans, highly talented kids, refugees, and so forth. We can follow the indirect definition of inclusion as:

"effective mechanisms to ensure the participation of students in public life, in re-solving issues that affect them directly" as well as "equal opportunities for students in need of special government care and for students in need of special government care" (Section 5) [26].

When looking more closely at the document, inclusion seems to be defined as the socialisation of students with SEN, giving them "accessibility and quality (. . .) regardless age, retardation or anything else" [26].

As can be seen, these aforementioned excerpts demonstrate that vulnerable categories of students, including those with SEN, seem to be recognised not only on an educational level, but also on social and cultural levels. Students with SEN have talents and the right to develop them, as is legally written down in the Strategy [26].

Finally, the Strategy [26] shows that inclusion is also an "intensification of work to eliminate various barriers in the framework of the implementation of the State Program of the Russian Federation "Accessible Environment" for 2011–2015" [27] This program was developed and approved by the Regulation of the Government of Russian Federation on 17 March 2011. The program provides the implementation measures to ensure unhindered access to priority facilities and services in the spheres of life of all individuals (at all ages) with SEN and other low-mobility groups of the population. Basically, the aforementioned State Program (Accessible Environment) started with an assurance of physical access for areas that individuals with SEN and from low-mobility group need to reach (sounding traffic lights and pedestrian crossings, ramps, wide doors, elevators etc.). It has been prolonged thrice so far, in 2015, in 2018, and in 2021.

Moving to the Federal Law of Education (FLE) [10], our analysis shows that this document does not directly mention the term inclusion even once.

In summary, the documents illustrate that expressions of the concept of inclusion narrowly focus on the correction of students requiring SEN assistance, and then assimilation of those to the exiting forms of general educational institutions. However, the expression of inclusion as a right for *all* seems to indicate movement towards a process by which the government endeavours to give accessibility to education for *all* learners.

### 5.2. The Inconsistent Expression and Definition of Inclusive Education

The second theme focuses on the definition and expression of the term inclusive education. Our analysis shows that neither FSES for Primary General Education [30] nor FSES for the Education of Students with ID [31] mention the term inclusive education. At the same time, these documents are intended to present the main provisions of inclusive

education and can be used by parents, education bodies as well as medical institutions as practical guides helping to adapt education for all through the various programs which are elaborated upon for educating students with various types of SEN: "[document- the practical instrument] is a set of mandatory requirements for the implementation of adaptive basic general education programs of general education" [30].

Additionally, the FSES for Students with ID [31] is directed towards the provision of education of special groups of students with ID, thereby labelling these as a separate group. This document sheds light on the opportunities and responsibilities of parents of students with ID as well as the conditions and criteria for the education of those students, without mentioning directly the implementation of inclusive education. The FSES for Students with ID, however, takes into account the age, typological and individual characteristics of students when planning the process of their education [31].

The findings demonstrate further that the Strategy [26] mentions the term inclusive education five times. Inclusive education is mentioned once regarding preschool education, twice in terms of students' rights to inclusive education, and finally, in order to create a supportive financial mechanism to protect the inclusive education of students with SEN. However, no clear definition of inclusive education is given.

Nevertheless, we can follow indirectly the definition of inclusive education as narrowly focusing on the rights to education and support students with SEN:

"Measures aimed at state support for students with LHC provides all notes with words "inclusive education (...). Legislative consolidation of legal mechanisms for the implementation of the rights of students with SEN to be included in the existing educational environment at the level of preschool, general and vocational education (the right to inclusive education)" (the Strategy, Section 4) [26].

At the same time, Section 4 expands the definition, thus directly mentioning the term inclusive education. The document highlights the prohibition of the discrimination and assess to inclusive education. Still, it is unclear whether document means that inclusive education is the only form acceptable education in the future for the students with SEN, or it can also mean that this form of education will be one of many forms: "Introduce an effective mechanism to combat discrimination in access to education for students with LHC in case of violation of their right to inclusive education" (the Strategy, Section 4) [26].

Additionally, the findings show that the Strategy [26] clearly mentions a need to improve the professional competence of teaching staff in the field of education and to establish the organisation of a system for training and retraining specialists working with students with SEN, as a main strategy for inclusive education for students with SEN (the Strategy, Section 3.5) [26].

In the FLE [10], the findings show that Article 2 (para. 27) clearly defines several basic concepts that are important for the organisation of education. Among those definitions, the description of inclusive education is given for the first time in Russian legal history.

The FLE emphasises inclusive education as a right ensuring equal access to education for *all students*, taking into account the diversity of SEN and individual opportunities. Moreover, the FLE states that, in the Russian Federation, the quality of education as well as social inclusion is guaranteed, especially for those with SEN. However, inclusive education, given as a right to education, is still clearly described as a correction of students' disabilities:

"The necessary conditions are created for receiving, without discrimination, quality education by persons with LHC, for the correction of developmental disorders (...) as well as the social development of these persons, including through the organization of inclusive education for persons with disabilities" (Article 5, point 5, part 1) [10].

The work on the strength of the quality of the education for all is further specified in FLE (Article 11) [10] through expressing the need for the development of educational standards. The FLE highlights that the standards will provide the possibility of flexible change in the educational route, especially for those with SEN, taking into consideration students' personal results and features as well as through the conclusions of the Psychological, Medical and Pedagogical Commission and the opinions of parents (legal guardians).

Additionally, the FLE, in Article 5 [10] clarifies the implementation of inclusive education for students with SEN by recommending ways of planning, monitoring, and evaluating educational provision for children and adults as well as ensuring adequate teacher training, as the Strategy [26] states.

Inclusive education is also indirectly expressed as a way of organising the educational process for students with SEN. The documents allow for various methods of organisation, including the strict segregation of students with SEN: "the education of students with disabilities can be organized both together with other students, or in separate classes, groups or in separate organizations carrying out educational activities (Article 79 para 4) [10].

Inclusive education incorporates the development of the adaptive basic general education programs (Article 79 para 3) [10]. The FLE points out that adaptive educational programs should take into account the characteristics of students' development, individual capabilities and, if necessary, to provide the services of an assistant and other conditions without which it is impossible or difficult to master educational programs. As can be seen, these programs are directed to support inclusive education, especially for students with SEN.

In summary, the results reveal an inconsistent definition of inclusive education in the analysed documents. On the one hand, the documents emphasise the right to ensure equal access for students with SEN. On the other hand, the documents focus on the correction of disabilities for students with various SEN. However, the definition of the term provided in the FLE shows a slight tendency towards an extension of the definition of inclusive education as equal access to education for *all* students, not only for those with SEN.

## 6. Discussion and Conclusion

The current paper's starting point was to examine how inclusion and inclusive education for students with SEN is understood in legislative and strategic documents at the state level between 2012 and 2014. Several of the documents, among those, the FLE [10], surprisingly, do not provide any clear definition of inclusion, and if they indirectly do so, the interpretations of the concept narrowly focus on the correction of students requiring SEN, and 'assimilation' into the educational sector, while the educational environment remains mostly unchanged. It is, however, important to point out that a clear definition of inclusive education is embodied in the FLE [10]. Still, we can note that legislative and strategic documents seem to support the practice of institutionalisation and segregation of students with SEN.

Such an interpretation potentially draws a complicated picture of SEN as something negative that must be reduced in order to minimise the differences between normal and "special" students [39].

Our findings, to some extent, cohere with previously reported research confirming that such an understanding of inclusion and inclusive education is close to the idea of integration [1,11,12]. Therefore, this can give the impression that the realisation of inclusion and inclusive education in Russia does not obviously follow the fundamental shift with respect to the way in which equity in education for *all* is addressed. Looked at in this this way, it is clear that these results are to some extent in dissonance with Ainscow et al. [23], who suggest using the definitions of inclusion and inclusive education as a process concerned with the identification and removal of barriers to the presence, participation and achievement of *all* students, not only for those with SEN.

Another reason behind those results may stem from the fact that the concepts of inclusion and inclusive education can be confusing since they may mean different things to different people [1]. Our results, to some degree, are consistent with Rytova et al. [18] who point out that it might be challenging to reach agreed definitions of these concepts in legislative and strategic documents when not taking into consideration the peculiarities of regions and the complexities of interactions between the different regions of Russia. As such, it can be assumed that a multitude of interpretations for legislative and strategic documents in various regions may lay foundations for implications for achieving a more

equitable and holistic education system in the country as a whole. Still, this fact does not give us a reason to reject inclusion and inclusive education as a goal for human beings, in order to increase the sense of belonging, and of feeling valued, to a social group [39].

Broadening the perspective, it is important to say that, despite there being no model of inclusive education that suits every country, a correlation exists between the ideology predominant in a society and its approach to inclusion and inclusive education [8,40]. It is timely to repeat that in Russia, after many years behind the Iron Curtain, the agenda of inclusion for all has been very slowly carried forward. We can rightly conclude that Russian society is still reluctant to abandon the segregated solution [1,8]. As such, new concepts of inclusion and inclusive education have not yet received the necessary justification and clarification at the legal and legislative level [11,13,20].

That said, we want to direct readers' attention towards a positive developmental direction found in the analysed documents. According to results of this study, we observed a slight movement towards the development of concepts towards inclusion in its broadest sense, for all students, and towards providing them with participation and high-quality education. Moreover, the documents tend to offer both students with SEN and their parents the right and possibilities to choose and be heard. This means that the documents may contribute to the development of new social values, namely the principles of equality and formal democratic rights. As such, there is hope concerning the development of a new fundamental core where society supports diversity and where all students are equally valued [39].

As a conclusion, a number of suggestions can be made. First, it is necessary to reflect and define the concepts of inclusion and inclusive education more closely in current legislative and strategic documents. Ainscow [1] has pointed out that if stakeholders do not share a common idea of what inclusion and inclusive education means, the direction remains unclear and the question of how education systems can be reformed and how support can be provided for students with SEN within an inclusive context may be overlooked.

Second, clear coherence between the documents needs to be established for further development of inclusive educational environments. Third, the already existing legislative and strategic documents need to be moderated, refreshed, and expanded in line with the modern development of society. Finally, this paper reflects the Russian context, but there is a reason to believe that our findings may apply to a broader international context. The development of inclusion and inclusive education is high on the international policy agenda. As such, our paper illustrates the power of using findings and the discussion of unusual contexts to help readers to reconsider policy and practice in readers' own contexts. In this way, challenges and possibilities may become clearer and become catalysts for new scrutiny and innovation [1].

**Author Contributions:** Conceptualization, N.B.H. and A.A.A.; methodology, N.B.H. and A.A.A.; software, N.B.H. and A.A.A.; validation, N.B.H. and A.A.A., formal analysis, N.B.H. and A.A.A.; investigation, N.B.H. and A.A.A.; resources, N.B.H. and A.A.A.; data curation, N.B.H. and A.A.A.; writing—original draft preparation, N.B.H. and A.A.A.; writing—review and editing, N.B.H. and A.A.A.; visualization, N.B.H. and A.A.A. All authors have read and agreed to the published version of the manuscript.

**Funding:** This research received no external funding.

**Institutional Review Board Statement:** Not applicable.

**Informed Consent Statement:** Not applicable.

**Data Availability Statement:** The data (Legislative and Strategic Documents) are available by contacting the corresponding author.

**Acknowledgments:** We would like to thank Elena N. Kutepova, E-mail: enkutepova@mail.ru, https://orcid.org/0000-0002-5347-9583, Center of Healing Pedagogics "Special Childhood"; Ph.D. in Education, Moscow, Russia for helping with data gathering and further remarks of the study. Without your contribution this paper would not have been published.

**Conflicts of Interest:** The authors declare no conflict of interest.

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
