# Peer review of "Inclusion and Inclusive Education in Russia: Analysis of Legislative and Strategic Documents at the State Level between 2012–2014"

_education, doi:10.3390/educsci14030312_

Round 1

Reviewer 1 Report

Comments and Suggestions for Authors

The theoretical framework is adequate, well grounded in the literature, even arguing the cause of failure to achieve the goals of inclusion, "misunderstanding and lack of acceptance of students with SEN by teachers and society in general", which is very relevant.

The diversity of development and funding of Inclusive Education in different Russian regions is highlighted. Dissemination of this work is useful to raise awareness of the situation and mitigate with measures to promote inclusion in different regions.

The Method section is well presented, however it is recommended to add a graph explaining the steps taken and at least some categories or codes that have been reached, in the Results section.

It is a very relevant and necessary work since, according to the authors' analysis, "legislative and strategic documents seem to support the practice of institutionalization and segregation of students with SEN". And this fact must be known in order to elaborate measures to correct it.

Recent and appropriate bibliography is used and fundamental authors in the field are studied.

The work gives rise to other researches such as: What do the concepts of inclusion and inclusive education mean for different people, professionals and groups, what to do to improve the understanding and acceptance of society, etc., and what is more important, to analyze the difference between what is stated in the legislation and what really happens in the educational centers.

Author Response

Thank you much for your valuable comments. the responce to reviewers is enclosed. 

Reviewer 2 Report

Comments and Suggestions for Authors

This manuscript reports the findings of a qualitative content analysis of four documents from the Federal State Educational Standard (FSES) in Russia, examining the conceptualization of inclusion and inclusive education for students with special education needs (SEN) in educational policy from 2012 to 2014. Through the analysis, the authors aim to shed light on the understanding of inclusion within the Russian education system during this time period.

Exploring how other nations include and implement inclusive education in their legislative documents would be beneficial, but the methods and practical implications of this study's findings require further explanation. I suggest that the authors take into consideration the following suggestions and comments.

Introduction

p 4, lines 167-72: It would be more beneficial to have a specific percentage of students who receive education at home and students at special boarding schools, in relation to the overall number of students or those with special educational needs. It is challenging to fully comprehend the situation with only the numbers (i.e., 40,000 and 70,000).

Methods

p 4, line 177- & p 6, line 190-: It may be beneficial to include subtitles within the Methods section. Potential options could include "Document Selection: Inclusion and Exclusion Criteria" starting on page 4, line 177, and "Analysis" starting on page 6, line 190.

p4, line 177-188: It may be beneficial for the authors to provide a more detailed explanation of the process used to search for and identify the Legislative and Strategic Documents. This could include information on how the initial search was conducted, who was involved in the screening process, and whether any assistants were utilized and trained. In addition, the inclusion of a table or figure, such as a PRISMA chart, could effectively demonstrate the number of initial documents selected, the screening process, and ultimately, the final number of documents selected for inclusion.

Were there any exclusion criteria included? Only inclusion criteria were mentioned.

P 6, line 190- p 7, line 229: The abstract mentioned deductive approach for the qualiative content analysis. Readers can discern that the manuscript utilized a deductive approach; nevertheless, explicitly mentioning the use of a "deductive approach" in the analysis section would enhance clarity. Additionally, it would be beneficial to explain the procedures of any "a priori codes or categories" that were developed from the theoretical perspective, providing insight into the concepts of inclusion and inclusive education.

P 7, line 230- p8, line 276: In my opinion, restructuring the placement of the 'Limitations' section to be between the Discussion and Conclusion would be more effective.

Results

p 8, line 287: The term FSES (Federal State Educational Standard) should be written out in full when it is first mentioned in the text. ID (Intellectual Disabilities) also needs to be written out in full.

P 8, line 297: There is no appendix attached in the manuscript.

P 10, line 394-398: I wonder if there is any specific legislation that explicitly states that inclusive education is the only form acceptable for students with SEN in the future. Many countries have laws and policies that promote and prioritize inclusive education as the preferred form of education for students with SEN. If authors have any exemplary legislation, it would greatly benefit readers if they could share the details of it.

Discussion

It is highly beneficial to gain a deeper understanding of how inclusive education is implemented within schools based on these Federal State Educational Standards (FSES).

It is understandable that finding and discussing unusual context can assist readers in re-evaluating policy and practice in their own setting. However, it would be more effective when data of current inclusive education practice is provided. Unfortunately, it is disheartening to note that there are schools in the US where students with SEN are kept completely apart from their peers without SEN, housed in separate buildings within the same school.

Author Response

Thank you very much for your valuable comments. The responce to reviewers is enclosed 

Round 2

Reviewer 2 Report

Comments and Suggestions for Authors

1. Please provide the proportion for the 729,000 students in the following sentence (page 4, line 169) and correct numbers (i.e., replace the decimal point with a comma to separate the thousands).

“According to the Federal State Statistics Service (Rosstat), there were 729.000 students with SEN under age 18 in 2019. This number includes preschool and school ages. Still, 40.000 students with SEN are educated at home (5.49% out of all students with SEN), and about 70.000 (9.6% out of all students with SEN) are educated at special or boarding schools, being limited to learning alongside their peers in local community schools.”

Here is the reason why I strongly suggest providing the proportion.

I acknowledge the importance and urgency of having strong policies and ongoing efforts to attain complete inclusive education. However, it might be necessary for readers to also recognize the value of clearly defining the terms "inclusive education" and "inclusive education" in legislative and strategic documents within the specific Russian context.

In 2018, Russia had a low percentage of students identified as students with SEN at only 2.40%, according to a recent study by Entrich (2021) (see https://eric.ed.gov/?id=EJ1291457). In comparison, the United States had a much higher proportion of students with SEN at 14.10% during the same time period (including more than 30% of Specific Learning Disabilities within the students with SEN population).

The relatively low percentage of students with SEN in Russia, compared to other countries, possibly indicates a system that only identifies students with the most significant support needs as having SEN. This also could suggest that the number of students with SEN in inclusive education could appear low if only those with severe support needs were identified as having SEN (although I prefer full inclusive education). If this is the case, it is highly probable that a student with SEN who is not included in general education in Russia would also face exclusion in other countries. In fact, the percentage of students with SEN who were not in inclusive settings as a portion of the overall student population was higher in other countries like Germany (3.08%), Denmark (4.87%), and the Netherlands (2.07%), than in Russia (1.75%).

The proportion of students with SEN compared to the total student population prompts readers to consider these critical issues. Moreover, it is also important to consider the varying perceptions of the term 'inclusion' in different cultural contexts such as Russia. For example, utilizing a term such as ‘Anti-discrimination’ in the legislative documents could potentially be more impactful and conducive to the success of inclusive education in a different culture and context (although the term inclusion is generally encouraged). Readers should be afforded the chance to reflect on these matters.

2. page 8, line 296-297: put ‘(FSES)’ after ‘Federal State Educational Standard’ (i.e., Federal State Educational Standard (FSES) for the Education).

Author Response

The authors would like to thank the reviewers for the thorough and meticulous reviews. Not only did the reviewers highlight the core meaning and methods, but also mentioned the preferred style and ethics of the article. We strongly believe that any article gets better and more helpful for the future researches with every new review.

1. Thank you so much for your opinion. We used periods for numbers to separate the thousands because generally both symbols (periods and commas) are the acceptable choice. However, period is more common in the United States rather than in Europe, and with regards to your point of view, we are more than happy to change periods to commas.

We understand and value your explanation, and we added the proportion. However, we would also like to add that Russia has just started its inclusive journey. In 2012, when the first legislative documents on the State level were established, Europe and the US have already been studying the phenomena of inclusion for more than 40 years. Thus the numbers in the mentioned countries are supposed to be more accurate than those we receive about Russia. We agree, though, that it would be indeed important to pay international attention to the varying perceptions of the term 'inclusion' in different cultural contexts such as Russia’s.

  1. Thank you for your attentiveness, we changed the mentioned lines for the better understanding.